# Digital Devices Use and Language Skills in Children between 8 and 36 Month

**DOI:** 10.3390/brainsci10090656

**Published:** 2020-09-21

**Authors:** Francesca Felicia Operto, Grazia Maria Giovanna Pastorino, Jessyka Marciano, Valeria de Simone, Anna Pia Volini, Miriam Olivieri, Roberto Buonaiuto, Luigi Vetri, Andrea Viggiano, Giangennaro Coppola

**Affiliations:** 1Child Neuropsychiatry Unit, Department of Medicine, Surgery and Dentistry, University of Salerno, Via Salvator Allende, 84081 Baronissi (Salerno), Italy; opertofrancesca@gmail.com (F.F.O.); valeriades@hotmail.it (V.d.S.); annapia.volini@tiscali.it (A.P.V.); mir.olivieri@gmail.com (M.O.); buonaiuto.roberto@libero.it (R.B.); aviggiano@unisa.it (A.V.); gcoppola@unisa.it (G.C.); 2Department of Mental Health, Physical and Preventive Medicine, Clinic of Child and Adolescent Neuropsychiatry, University of Campania “Luigi Vanvitelli”, 80138 Naples, Italy; 3Department of Neurosciences, Imaging and Clinical Sciences, University of Chieti “G. D’Annunzio”, 66100 Chieti, Italy; jmarciano.psy@libero.it; 4Department of Experimental Psychology, University of Oxford, Oxford OX2 6GG, UK; 5Department of Health Promotion, Mother and Child Care, Internal Medicine and Medical Specialties (PROMISE), University of Palermo, 90127 Palermo, Italy; luigi.vetri@gmail.com

**Keywords:** digital devices, digital media, toddler, children, language abilities

## Abstract

**Background**: Over the past decade, the use of digital tools has grown and research evidence suggests that traditional media and new media offer both benefits and health risks for young children. The abilities to understand and use language represent two of the most important competencies developed during the first 3 years of life through the interaction of the child with people, objects, events, and other environmental factors. The main goal of our study is to evaluate the relationship between digital devices use and language abilities in children between 8 and 36 month, also considering the influence of several factors. **Materials and Methods**: We conducted a cross-sectional observational study on digital devices use and language abilities in260 children (140 males = 54%) aged between 8 and 36 months (mean = 23.5 ± 7.18 months). All the parents completed a self-report questionnaire investigating the use of digital devices by their children, and a standardized questionnaire for the assessment of language skills (MacArthur-Bates). Linear regression analysis was used to evaluate the relation between different variables. Subsequent moderation analysis were performed to verify the influence of other factors. **Results**: We found a statistically significant negative relation between the total daily time of exposure to digital devices and the Actions and Gestures Quotient (ß = −0.397) in children between 8 and 17 months, and between the total daily time of exposure to digital devices and Lexical Quotient (ß = −0.224) in children between 18 and 36 months. Gender, level of education/job of parents, modality of use/content of digital device did not significantly affect the result of the regression analysis. **Conclusion**: In our study we found that a longer time of exposure to digital devices was related to lower mimic-gestural skills in children from 8–17 months and to lower language skills in children between 18 and 36 months, regardless of age, gender, socio-economic status, content, and modality of use. Further studies are needed to confirm and better understand this relation, but parents and pediatricians are advised to limit the use of digital devices by children and encourage the social interaction to support the learning of language and communication skills in this age group.

## 1. Introduction

The abilities to understand and use language represent two of the most important competencies developed during the first 3 years of life [1,2,3,4,5] through the interaction of the child with people, objects, events, and other environmental factors in everyday settings [6,7,8]. This time is a great period of plasticity in which the brain is sensitive to the quantity and quality of the linguistic input heard from both people and object input sources. In fact, children’s language skills rapidly unfold over the first three years of life [9] and during this time, language acquisition is determined by the total number of words heard in children’s everyday environments and by the syntactic richness and complexity of language expressed in the home environment [10].

Infants and toddlers are also capable of learning from screen media. This learning is dependent upon the confluence of three distinct but interrelated factors: attributes of the child; characteristics of the screen media stimuli; and the varied environmental contexts surrounding the child’s screen media use [5].

In the past years, the market for infant-directed media products has increased and researchers are trying to understanding whether and how screen media impacts language development.

This increased use reflects both the increasing use of screen media by families and society [11] and the growing marketing of cable TV channels, digital devices, and applications (apps) to young children [12] even to those from disadvantaged households [13].

There is considerable evidence that children under two who watch educational television do learn media-presented vocabulary and then are able to transfer specific learning to more generalized language gains [5,14,15,16,17,18,19] but specifically, Linebarger and Walker (2005) [20] find that outcomes were program specific. Programs that had a strong narrative, such as Dora the Explorer, were positively associated with greater vocabulary and expressive language, whereas, programs that had little narrative structure and spoken language, such as Teletubbies, were negatively associated with vocabulary and expressive language.

This suggest that not only does the amount of television exposure influences language development, but also there are other important factors to consider. A determinant role in the screen exposure is played by parents. Parents can offset some of the potential harmful effects of media exposure on their children for example stimulating an active use rather than a passive one.

However, it would appear that the quality and quantity of parents’ interactions with their children tend to be reduced by the presence of television [21,22,23].

Kabaliet al. (2015) [13], in a study conducted in a low-income urban pediatric clinic, showed that almost all (97%) 0 to 4 year olds had used a mobile device, and three-quarters owned their own device. The 92% of 1 year olds had exposure to mobile devices and they were primarily using mobile devices for entertainment, not educational, purposes. Many parents report that mobile devices, which are handheld and usually used individually, are more difficult to monitor in terms of what the child is playing or downloading as well as where and when they are using media. Instant accessibility means that children can demand preferred programs at any time or place and Hinikeret al. (2016) [24] found that context-based rules (i.e., where children are allowed to use digital media, such as the dinner table) were the hardest to enforce compared with rules about time limits and content. Mobile devices are used to placate or distract children or to manage children’s behavior. Studies revealed that parents often give children devices when doing house chores, to keep them calm in public places, during meals and/or at bedtime to put their child to sleep [13,25,26,27,28].

In Italy a recent survey described that20% of children used a smartphone for the first time during his first year of life. Moreover, 80% of children from 3 to 5 years old is able to use their parent’s Smartphone. In addition, parents often use media as pacifiers, giving mobile devices to their child to keep them calm during the first (30%) and the second (70%) year of life [29,30]. The majority of the studies in the literature are on the US population but there are very few studies on Italian infants, toddlers, and preschoolers. For that reason we believe it is important to explore how mobile devices are managed by parents and how much this affects language development in Italian children under three years old.

The main goal of our study was to explore the relationship between the language abilities and the use of digital devices (DD) in children between 8 and36 months, assuming that the use of DD adversely affects language development.

The secondary objective of the study was to evaluate the influence of other factors on this relationship. Based on previous literature data, the following variables were selected: gender, socio-economic status, co-viewing, contents of DD, frequency of social activities [5,6,9,10,11]. We hypothesize that co-viewing and type of content of DD influence the relationship between DD use and language development. In particular, we consider that there is a difference between children that use devices in active interaction with parent and children that use devices alone. We do not expect socioeconomic status differences as each child has full access to devices.

## 2. Materials and Methods

### 2.1. Sample Selection and Characteristics

We conducted a cross-sectional observational study on digital devices use and language abilities in 260 children (140 males = 54%; 120 female = 46%) aged between 8 and 36 months (mean = 23.5 ± 7.18 months), recruited from twelve kindergartens and nursery schools of the city of Salerno.

All the children younger than or equal to 36 months were included; the exclusion criteria were the presence of medical or neuropsychiatric conditions that could affect language or neuropsychomotor development, and poor parental compliance to take part in the study.

All the parents were invited to a preliminary meeting with a specialist in Child Neuropsychiatry, in order to explain the methods and purposes of our research.

Subsequently, the following self-report questionnaires were administered to the parents who provided their written informed consent:
−Digital Devices Questionnaire (DDQ): a non-standardized questionnaire that aimed to investigate the use of DD by the children;−Il primo vocabolario del bambino PVB “Gesti e Parole” Forma Breve—Italian adaptation of the MacArthur-Bates Communicative Development Inventory—CDI “Actions and Gestures” Short Form: a standardized questionnaire for the assessment of language abilities of children aged between 8 and 17 months;−Il primo vocabolario del bambino PVB “Parole e Frasi” Forma Breve—Italian adaptation of the MacArthur-Bates Communicative Development Inventory—CDI Short Form “Actions and Gestures” Short Form: a standardized questionnaire for the assessment of language abilities of children aged between 18 and 36 months.

All the data were collected and examined by a single investigating neuropsychiatrist.

The sample was divided into two sub-groups: 72 children between 8 and 17 months (mean = 13.8 ± 3.5) and 188 children between 18 and 36 months (mean = 27.1 ± 4.4), based on the typical differences in language competences of the two group age and on the different standardized questionnaires administered. Socio-demographic and clinical data suggest that the sample is representative of a population with a typical psychomotor development (first steps mean age = 12.6 ± 2.0 months; first words mean age = 12.8 ± 4.2 months) and the two subgroups appeared homogeneous for the main characteristics analyzed. The socio-demographic characteristics of the sample are summarized in Table 1.

This study was performed in accordance with the Helsinki declaration and was approved by the Campania Sud Ethics Committee (protocol number = 0033986; det. N. 32–05 March 2019).

### 2.2. Digital Devices Questionnaire (DDQ)

Digital Devices Questionnaire consisted of two parts:

A first section that collected general information, socio-demographic data, and medical history (age, gender, family members, number of siblings, educational level and job of the parents, pregnancy, childbirth, psychomotor development, diagnosed pathologies); a second section, consisting of 12 items, exploring the use of DD by children, as follows:
−DD available at home (smartphone, tablet, personal computer, television, videogames);−children’s favorite DD (smartphone, tablet, personal computer, television, videogames);−age of start using DD (smartphone, tablet, personal computer, television, videogames);−mean time of daily use (smartphone, tablet, personal computer, television, videogames);−modality of use (with or without parents’ supervision);−preferred content (with or without dialogues);−content selection (independent choice or choice guided by the parent);−behaviors implemented by DD use (frustration level, name response, social attention);−parental motivation for allowing DD to the child (to entertain, to calm down, during meal time, to let the child sleep);−parents’ awareness of the risks for their children associated with the DD use;−parents request to the pediatrician for advice on DD use by children;−time spent by the child in social activities with peer (times in which the child plays or interacts with other children).

### 2.3. Il Primo Vocabolario del Bambino (PVB) Forma Breve - Italian Adaptation of the MacArthur-Bates Communicative Development Inventory (CDI) Short Form

The language skills of the children were assessed through a standardized questionnaire.

PVB is a standardized questionnaire for parents of children aged between 8 and 36 months, used both in research and in clinical practice for assessing communication and language in children with typical and atypical development. Given the physiological changes in language development between the first and third year of age, two forms have been created: the “Gestures and Words” Form, for children between 8 and 17 months, and the “Words and Phrases” Form, for children between 18 and 36 months.

All the raw scores, are converted into scores standardized for age.

In the “Gestures and Words” Form, three standardized scores are considered:
−Actions and Gestures Quotient (AGQ): parameter that evaluates the mimic-gestural communication skills.−Lexical Understanding Quotient (LUQ): parameter that evaluates the comprehension of words.−Lexical Production Quotient (LPQ):parameter that evaluates the production of words.

In the “Words and Phrases” Form, one standardized score is considered:
−Lexical Quotient (LQ): parameter that evaluates the production of words.

The standardized scores have mean = 100 and standard deviation (SD) = 15; scores below 70 (−2 SD) are considered below the norm.

### 2.4. Statistical Analysis

All data were expressed as mean, standard deviation and proportions/percentage, and subjected to descriptive statistics analysis.

A preliminary normality test was performed in order to verify the data distribution (Shapiro-Wilk Normality Test). Linear regression analysis was used to evaluate the relation between different variables. Subsequent moderation analysis were performed to verify the influence of other factors (gender, parents’ educational level, parent’s job, co-viewing, contents of DD, frequency of social activities). Bonferroni correction for multiple comparisons has been applied; *p*-values less than or equal to 0.0025 were considered statistically significant. All data were analyzed using the Statistical Package for Social Science, version 23.0 (IBM Corp, 2015, Armonk, NY, USA).

## 3. Results

### 3.1. Use of Digital Devices by Children

Most of the parents reported the use of at least one DD by their children (252/260, 97%); in younger children this percentage was slightly lower (64/72, 89%), while it reached 100% in the group of older children (188/188).

The DD preferred by children in both groups were the smartphone (171/252, 66%) and the television (157/252, 60%), with a slight preference for the smartphone in older than in younger children (71% vs. 53%). The tablet ranked third (44/252,17%), with a greater preference in older children (21% vs. 6%).

The parents reported that the DD used as first by the children were television (mean age = 11.3 ± 5.2 months) and smartphone (15.6 ±5.8 months), followed by tablet (18.5 ± 6.1 months), personal computer (18.8 ± 7.9 months), and videogames (22.3 ± 3.2 months). In the group of younger children, the mean age of the first use of any DD considered was always earlier, compared to the group of older children (smartphone: 11.1 ± 3.1 vs. 16.4 ± 5.9; tablet: 15.3 ± 3.1 vs. 18.6 ± 6.2; personal computer: 14.0 ± 3.1 vs. 16.8 ± 12.9; television: 8.4 ± 3.7 vs. 12.2 ± 5.3; videogames: 16.0 ± 0.0 vs. 23.6 ± 1.3).

In the total sample, the mean time spent on DD by the children was 2.13 ± 2.04 h/day. The most used DD were once more television (1.27 ± 1.39 h/day) and smartphone (0.62 ± 0.85 h/day). Older children used DD more than younger (2.34 ± 2.09 vs. 1.57 ± 1.79 h/day).

Most of the children used DD mainly in presence of parents/caregivers (188/252, 75%), or of brother/sister (23/252, 9%), while a 5% of children used DD without any supervision (13/188); a scheduled digital parent-control was reported in 11% (28/252).

The children’s favorite DD content was video with dialogues (190/252, 75%). More in detail the content of DD were driven by the parent in 89% of younger children (57/64, 89%), while only in 41% (110/188) of older one.

In both groups the main reasons why parents allowed their children to use DD were “to entertain” (104/252, 42%) or “to calm the child” (91/252, 36%). The DD were frequently also used during meal time (94/252, 37%) or before the child went to sleep (23/252, 9%).

In few children, with a slight prevalence in the older group, the parents reported that they did not respond to the their name when called (6/252, 2%), or did not interact with others (12/252, 5%) during DD use. A further 6% (16/252) appeared frustrated, with stubborn crying, if the DD was taken away.

Overall, the parents also reported sleep problems (including difficulty falling asleep and/or frequent nighttime awakenings) in about 33% (86/260) of the children, with a slight prevalence in younger group(45% vs. 33%).

Finally, 53% of parents (134/260) expressed concern about the health consequences of the DD use in their children, although only 19% of the parents (47/260) had already asked their pediatrician for advice on this topic. All data concerning the use of DD by children are summarized in Table 2.

### 3.2. Relation between Digital Devices Use and Language Skills in Children

In the group of younger children, a statistically significant negative relation was found between the total daily time of exposure to DD and the Actions and Gestures Quotient (AGQ) scores, through linear regression analysis (ß = −0.397; R^2^ = 0.158; *p* = 0.001; Table 3; Figure 1).

Subsequently, moderation analysis showed that other factors including gender, level of education and job of parents, co-viewing, modality of DD use, and frequency of social activities did not have a significant influence on the result of the regression analysis (Table 4).

On the other hand, no statistically significant relation was found between the total daily time of exposure to DD and the Lexical Understanding Quotient (LUQ) and Lexical Production Quotients (LPQ) scores (Table 3).

The relation between age of the first use of the DD and AGQ, LUQ and LPQ scores was also tested, but no statistically significant relation was found (Table 3).

In the group of older children a statistically significant negative relation was found between the total daily time of exposure to DD and the Lexical Quotient (LQ) scores, through linear regression analysis (ß = −0.224; R^2^ = 0.060; *p* = 0.001; Table 3; Figure 2).

Also in this case, we found that the variables considered in the subsequent moderation analysis did not significantly influence the results of the regression analysis (Table 4).

No statistically significant relation was found between the start age of use of the DD and the LQ scores (Table 3).

## 4. Discussion

The purpose of our study was to evaluate the use of digital devices in a population of children under 3 years old and relate them with language skills. Our sample consisted of 260 children (54% male; mean age = 23.5 ± 7.2 months), that was divided into two subsamples of children aged 8–17 months (*n* = 72) and 18–36 months (*n* = 188), according to the physiological variations of language skills in the two different age groups. The two subgroups appeared homogeneous for the main socio-demographic and clinical characteristics analyzed, and both are representative of a typically developing population (the stages of psychomotor development were described in the norm and children with neuropsychiatric problems were not reported).

The first important result that emerged from our survey was that 97% of children used at least one of the most popular digital device, with a slight difference based on age. In particular, all the children between 18 and 36 months (100%) used DD against 89% of the children between 8 and 17 months. Our result is in line with the study of Zimmerman et al. (2007) [31] in which 90% of parent reported that their children younger than 24 months use some form of electronic media.

The digital devices preferred by the children were the smartphone (66%) and the television (60%), with a slight preference for the smartphone in the older children group than in the younger children group (71% vs. 53%). The tablet was in third place being preferred by 17% of the total sample and by 21% of children between 18 and 36 months.

This data confirm that the smartphone and the tablet, considered among “new” digital devices, have become part of everyday life, and that children, even very young, are increasingly familiar with these tools [32].

In addition, parents reported that the first digital device used by the children was television (mean age of first use = 11.3 ± 5.2 months) followed by smartphone (15.6 ± 5.8 months), tablet (18.5 ± 6.1 months), personal computer (18.8 ± 7.9 months), and videogames (22.3± 3.2 months). In our opinion, it is also important to underline that in the group of younger children the ages of first use of DD are always lower than in the group of older children, for all the digital devices analyzed. This data would suggest a sort of “anticipation” of the age to DD exposure, which would seem increasingly early [31,32].

The average daily used of DD was about 2 h/day, with a slight difference between the two groups (1.57 h in children between 8 and 17 months and 2.34 h in children between 18 and 36 months), confirming what has already emerged from the previous literature data. Vandewater et al. (2007) reported that forty percent of children between 6 and 23 months used digital media 2 or more hours/day [33] and Zimmerman et al. (2007) reported that on average infants younger than 24 months watched television for 1–2 h/day [31].

Most of the children used digital devices in the presence of the parent/caregiver (75%) or siblings (9%), and only a minority of children completely alone (5%); 11% of parents report the use of automatic parental-control.

In line with previous literature data, we also found that parents using DD as peacekeeper for their children while they are engaged in other activities [34,35]; in both groups the main reasons why parents allowed their children to use DD were to entertain (42%), to calm the child (36%), during meal time (37%), or before the child went to sleep (9%).

In a small percentage of children aged 18–36 months, parents reported some atypical behaviors during the DD use: These children did not respond to their name when called (2%), or did not interact with others (5%) during DD use; 6% of the children appeared frustrated, with stubborn crying, if the DD was taken away. This data suggest that the use of digital devices in some children can lead to a reduction in social interaction and a difficulty in emotional regulation, as already reported in previous studies [36,37].

Overall, the parents also reported sleep problems (including difficulty falling asleep and/or frequent nighttime awakenings) in about 33% of the children, with a slight prevalence in younger group (45% vs. 33%). The association between sleep disturbance and excessive media use by children had already been reported. In particular, a recent cross-sectional study including 1117 toddlers showed that everyday use of a tablet or smartphone raised the odds ratio of a shorter total sleep time and a longer sleep onset latency regardless of other factors, such as temperament or type of screen exposure (TV or Videogames) [38].

Finally, 53% of parents expressed concern about the health consequences of the DD use in their children, although only 19% of the parents had already asked their pediatrician for advice on this topic.

The most important result of the study concerned the relation between language skills and time spent on digital devices by the children.

In children aged 8–17 months, we found a negative relation between the total daily time of exposure to DD and the Actions and Gestures Quotient scores. Therefore, the children who spent more time to use digital devices showed a repertoire less rich in communicative gestures. It is important to consider that gestural and mimic ability are the main indicators of the pre-verbal communication skills of children in this age group.

In children aged 18–36 months, we found a negative relation between the Lexical Quotient and the time spent on DD. In this case, greater use of DD by children was associated with less production of words. In both cases the relation found was weak but significant and gender, age, and socio-economic status did not significantly affect these relations.

For several years there has been a scientific debate about the association between the use of digital media and language skills in children. In terms of quantity of exposure, many authors suggest that the use of digital devices may represent a “passive” behavior that displaces fundamental learning opportunities for the child [37]. Population-based studies and a very recent meta-analysis showed associations between excessive TV viewing in early childhood and language, cognitive and socio-emotional delay; possible mechanisms responsible for this association would be inappropriate content, decrease in parent–child interaction, and poor family functioning. Furthermore, earlier age of media use onset, cumulative hours of media use and contents were independent predictors of poor cognitive and linguistic skills [31,39,40,41,42].

In particular, evidence provides limited educational benefits of media use for children under 2 years and the American Academy of Pediatric discourage media exposure under this age [43].

It would seem, indeed, that before the age of 24 months, interaction with parents/caregivers is more effective in teaching verbal and non-verbal problem solving strategies [44]. On the other hand, there would be a difficulty learning from 2D representations before the 30 months (video deficit) due to the poverty of symbolic thought, control of immature attention and insufficient flexibility to transfer knowledge to the real world [45]. Before the age of two, therefore, children would learn language, sensorimotor, and socio-emotional skills more through hand-on exploration and social interaction.

Our study is in line with this evidence, showing that a higher use of digital media was associated with less language skills; however, some authors disconfirm these results, showing an absence of relationship between language skills and time spent by children on digital media [46]. Other studies suggest, on the other hand, that the use of digital devices is not exempt from bringing benefits to children, even if they depend on the age, the stage of development, the characteristics of the child, the methods of use (co-viewing) and content (educational applications). The use of educational applications and parent co-viewing would be associated with an improvement in language skills in children [47,48].

In our study the co-viewing and the digital contents did not significantly affect the relation between language and digital device use. This result could be due to not specifically investigating the use of “educational applications” and not to distinguish between “active” or “passive” co-viewing; therefore we can further explore these two aspects in future researches. Another limitation of the study is to use questionnaires aimed at parents. Further studies would be needed to assess language and communication skills through standardized direct tests [49].

## 5. Conclusions

Over the past decade, the use of digital tools has grown and research evidence suggests that traditional media and new media offer both benefits and health risks for young children.

In our study we found that a longer time of use of digital devices was related to lower mimic-gestural skills in children from 8–17 months and to lower language skills in children between 18 and 36 months, regardless of age, gender, socio-economic status, content, and modality of use. Further studies are needed to confirm and better understand this relationship, but parents and pediatricians are advised to limit the use of digital devices by children and encourage the social interaction to support the learning of language and communication skills in this age group.

## Figures and Tables

**Figure 1 brainsci-10-00656-f001:**
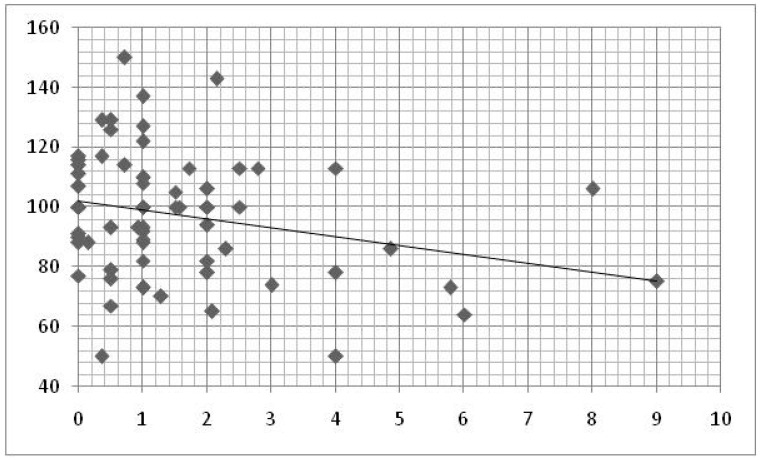
Linear regression between time of DD use by the children and *Actions and Gesture Quotient*. x axis = hours/day of DD use; y axis = Actions and Gestures Quotient scores.

**Figure 2 brainsci-10-00656-f002:**
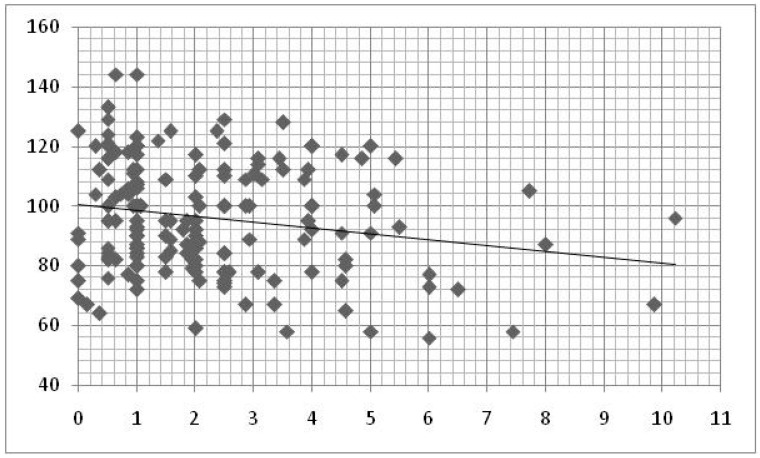
Linear regression between time of DD use by the children and *Lexical Quotient*. x axis = hours/day of DD use; y axis = Lexical Quotient scores.

**Table 1 brainsci-10-00656-t001:** Sample characteristics. m = mean; SD = standard deviation; M = mother; F = father; MS = middle school; HS = high school; UN = university; U/H = unemployed/housewife; SW = skilled worker; OW/T = office worker, teacher; SE = self-employed.

	Total Sample				8–17Months				18–36Months			
Sample Size	260				72				188			
Gender												
male	140 (54%)				44 (61%)				96 (51%)			
female	120 (46%)				28 (39%)				92 (49%)			
Age-m± SD	23.5 ± 7.2 months				13.8 ± 3.5				27.1 ± 4.4			
Pregnancy problems	38 (15%)				7 (10%)				31 (16%)			
Birth problems	27 (10%)				3 (4%)				24 (13%)			
Perinatal problems	25 (10%)				7 (10%)				18 (10%)			
Birth week-m± SD	38.9 ± 1.9				39.1 ± 1.4				38.8 ± 2.0			
Birth weight-m± SD	3.2 ± 0.5 kg				3.3 ± 0.5				3.2 ± 0.5			
Crawling	36 (14%)				10 (14%)				26 (14%)			
First step age m± SD	12.6 ± 2.0 months				12.3 ± 1.9				12.7 ± 2.0			
First word age m± SD	12.8 ± 4.2 months				10.8 ± 3.0				13.4 ± 4.3			
Medical pathologies	18 (7%)				4 (5%)				14 (7%)			
Family members	4.0 ± 1.0				3.7 ± 0.9				4.1 ± 1.2			
Siblings-m± SD	1.1 ± 0.9				1.0 ± 0.7				1.2 ± 0.9			
M/F age-m± SD	34.7 ± 5.1/37.7 ± 5.7 years				34.9 ± 4.7/37.3 ± 6.4				34.7 ± 5.3/37.8 ± 5.4			
Educational level	MS	HS		UN	MS	HS		UN	MS	HS		UN
Mother	27(11%)	116(45%)		114(44%)	4(6%)	28(39%)		39(55%)	23(12%)	88(47%)		75(41%)
Father	50(20%)	118(47%)		81(33%)	11(16%)	32(47%)		25(37%)	39(22%)	86(47%)		56(31%)
	U/H	SW	OW/T	SE	U/H	SW	OW/T	SE	U/H	SW	OW/T	SE
Mother	67(27%)	55(22%)	57(23%)	71(28%)	17(24%)	14(19%)	17(24%)	24(33%)	50(28%)	41(23%)	40(22%)	47(27%)
Father	14(6%)	87(35%)	70(28%)	77(31%)	2(3%)	25(36%)	15(22%)	27(39%)	12(7%)	62(34%)	55(31%)	50(28%)

**Table 2 brainsci-10-00656-t002:** Digital Devices Questionnaire; m = mean; SD = standard deviation; DD = digital devices; * (only for children ≥12 months, total sample size = 236).

	Total Sample			8–17 Months			18–36 Months		
Sample Size	260			72			188		
Use of DD by children (at least one)	252 (97%)			64 (89%)			188 (100%)		
DD available at home									
Smartphone	191 (74%)			43 (60%)			148 (79%)		
Tablet	86 (33%)			13 (18%)			73 (39%)		
Personal Computer	52 (20%)			19 (26%)			33 (18%)		
Television	230 (85%)			59 (82%)			171 (91%)		
Videogames/console	12 (5%)			2 (3%)			10 (5%)		
Children’s favorite DD									
Smartphone	171 (66%)			38 (53%)			133 (71%)		
Tablet	44 (17%)			4 (6%)			40 (21%)		
Personal Computer	12 (5%)			5 (7%)			7 (4%)		
Television	157 (60%)			40 (56%)			117 (62%)		
Videogames/console	4 (2%)			1 (1.4%)			3 (1.6%)		
Age of start using DD	months (m ± SD)			months (m ± SD)			months (m ± SD)		
Smartphone	15.6 ± 5.8			11.1 ±3.1			16.4 ± 5.9		
Tablet	18.5 ± 6.1			15.3 ± 3.1			18.6 ± 6.2		
Personal Computer	18.8 ± 7.9			14.0 ± 3.1			16.8 ± 12.9		
Television	11.3 ± 5.2			8.4 ± 3.7			12.2 ± 5.3		
Videogames/console	22.3 ± 3.2			16.0 ± 0.0			23.6 ± 1.3		
Time of use DD	hours/day (m ± SD)			hours/day (m ± SD)			hours/day (m ± SD)		
Smartphone	0.62 ± 0.85			0.41 ± 0.62			0.70 ± 0.91		
Tablet	0.19 ± 0.55			0.06 ± 0.23			0.25 ± 0.62		
Personal Computer	0.04 ± 0.28			0.05 ± 0.21			0.03 ± 0.30		
Videogames	0.01 ± 0.05			0.01 ± 0.06			0.01 ± 0.04		
Television	1.27 ± 1.39			1.04 ± 1.36			1.36 ± 1.39		
Total Time	2.13 ± 2.04			1.57 ± 1.79			2.34 ± 2.09		
Modality of use									
with parent/caregiver	188 (75%)			56 (87%)			132 (70%)		
with brother/sister	23 (9%)			4 (6%)			19 (10%)		
with parent-control	28 (11%)			3 (5%)			25 (13.5%)		
alone	13 (5%)			1 (2%)			12 (6.5%)		
Favorite contents									
with dialogue	190 (75%)			44 (69%)			146 (78%)		
without dialogue	22 (9%)			12 (19%)			10 (5%)		
Videogames	40 (16%)			8 (12%)			32 (17%)		
Contents selection									
independent choice	117 (46%)			7 (11%)			110 (59%)		
choice guided by parents	135 (54%)			57 (89%)			78 (41%)		
Reasons for granting DD									
to entertain	105 (42%)			24 (38%)			81 (43%)		
to calm down	91 (36%)			20 (31%)			71 (38%)		
to let the child sleep	23 (9%)			6 (9%)			17 (9%)		
during meal time	94 (37%)			24 (38%)			70 (37%)		
Concerns about DD use									
Yes	134 (53%)			38 (59%)			96 (51%)		
No	118 (47%)			26 (41%)			92 (49%)		
Request to the pediatrician									
Yes	47 (19%)			5 (8%)			42 (22%)		
No	205 (81%)			59 (92%)			146 (78%)		
Behavior during DD use									
response to name	absent	partial	immediate	absent	partial	immediate	absent	partial	immediate
	6 (2%)	92 (37%)	154 (61%)	0 (0%)	14 (22%)	50 (78%)	6 (3%)	78 (42%)	104 (55%)
social attention	absent	partial	adequate	absent	partial	adequate	absent	partial	adequate
	12(5%)	30 (12%)	210 (84%)	4 (6%)	4 (6%)	56 (88%)	8 (4%)	26 (14%)	154 (82%)
frustration	high	medium	low	high	medium	low	high	medium	low
	16 (6%)	113 (45%)	129 (51%)	4 (6%)	16 (25%)	44 (69%)	12 (6%)	99 (53%)	77 (41%)
Social activities * less than once a week	16 (7%)			4 (8%)			12 (6%)		
about once a week	72 (31%)			8 (17%)			56 (30%)		
several times a week	93 (39%)			16 (34%)			75 (40%)		
often- every day	55 (23%)			10 (21%)			45 (24%)		
Sleep problems	86 (33%)			32 (45%)			54 (29%)		

**Table 3 brainsci-10-00656-t003:** Linear regression analysis.

	Daily Time of Use DD				Start Age of Use DD			
	*R* ^2^	*ß*	*t*	*p*-Value	*R* ^2^	*ß*	*t*	*p*-Value
Lexical Understanding Quotient	0.033	−0.182	−1.515	0.135	0.053	−0.230	−1.495	0.143
Lexical Production Quotient	0.000	0.000	0.003	0.997	0.051	−0.225	−1.461	0.152
Actions and Gesture Quotient	0.158	−0.397	−3.542	0.001	0.034	−0.185	−1.192	0.240
Lexical Quotient	0.060	−0.244	−3.291	0.001	0.001	−0.032	−0.411	0.681

**Table 4 brainsci-10-00656-t004:** Moderation analysis.

	Actions and Gestures Quotient	Lexical Quotient
gender	*p =* 0.622	*p* = 0.528
parents’ educational level	*p* = 0.132	*p* = 0.251
parent’s job	*p* = 0.261	*p* = 0.475
co-viewing	*p* = 0.659	*p* = 0.728
contents	*p* = 0.969	*p* = 0.601
social activities	*p* = 0.403	*p* = 0.177

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
