# Peer review of "Digital Devices Use and Language Skills in Children between 8 and 36 Month"

_brainsci, 2020, doi:10.3390/brainsci10090656_

Round 1

Reviewer 1 Report

Overall this was a very carefully designed and implemented study. The presentation also is more than adequate.

I would suggest minor corrections which are mainly typos.

  1. Instead of "sex", I would suggest to use "gender"
  2. In line 63, change "...products is increased..." to "...products has increased..."
  3. There are in-text citations that follow the APA style (e.g. lines 70, 81, 90) and others are in bracets. Obviously this needs to be corrected
  4. Instead of "...and colleagues", I would keep the "...et al." for consistency.
  5. In lne 81 the last word should be "that" instead of "as", I think
  6. In line 98, make it "... in the literature ..." and the last word should be "infants"
  7. The authors use the DD abbreviation but they do not explain it the first time Digital Device is mentioned.
  8. In line 138 a "-" is missing at the beginning
  9. The tables are outside the page margins
  10. In the references, there is a double numbering.

Overall it is a good paper

Author Response

Salerno, 17 September 2020

To Editor-in-Chief

Brain Sciences

Dear Editor,

Thank you for considering our manuscript for publication in Brain Sciences.

We greatly appreciated the reviewer’s comments and suggestions. We did our best to address all the issues raised by reviewer. The manuscript has been modified, according to the reviewer’s comments and remarks. We have italicized the comments and addressed each of them in a point-by-point reply.

We hope that the manuscript is now acceptable for publication in Brain Sciences.

Looking forward to hearing from you,

Grazia M.G. Pastorino

COMMENTS TO THE AUTHORS

REVIEWER: 1

Reviewer comment:

Overall this was a very carefully designed and implemented study. The presentation also is more than adequate. I would suggest minor corrections which are mainly typos.

  1. Instead of "sex", I would suggest to use "gender"
  2. In line 63, change "...products is increased..." to "...products has increased..."
  3. There are in-text citations that follow the APA style (e.g. lines 70, 81, 90) and others are in bracets. Obviously this needs to be corrected
  4. Instead of "...and colleagues", I would keep the "...et al." for consistency.
  5. In lne 81 the last word should be "that" instead of "as", I think
  6. In line 98, make it "... in the literature ..." and the last word should be "infants"
  7. The authors use the DD abbreviation but they do not explain it the first time Digital Device is mentioned.
  8. In line 138 a "-" is missing at the beginning
  9. The tables are outside the page margins
  10. In the references, there is a double numbering.

Overall it is a good paper

Authors’ response:

We thank the reviewers for the comments and the suggestions. We made all required changes, as follows:

  1. We replaced the term “sex” with “gender” throughout the manuscript.
  2. We changed the sentence as suggested.
  3. We modified the citations in the text as suggested, making them uniform with the rest of the manuscript.
  4. We changed “and colleagues” in “et al.”.
  5. We replaced “as” with “that”.
  6. We modified the words as suggested.
  7. We made this change by adding the abbreviation DD after the first time we mentioned Digital Devices, at the end of the Introduction section, (line 102).
  8. We added “-“ as suggested.
  9. We added a new file with all tables.
  10. We removed the double numbering in the references.

Reviewer 2 Report

The study investigates the relationship between digital device use and language abilities in Italian children between the age of 8-36 months. The study’s significance would be an effort to provide valuable information on how the duration of using digital devices was related to children’s language skills. The study also provides a discussion of the negative effect of using digital devices in early years to parents and pediatricians. This study is interesting and significant in the field of language and literacy development of young children. The use of digital devices of young children is a very timely manner topic in this Pandemic situation since more digital devices are used in the education field. Thus, I believe that the findings reported in this study are relevant to the Brain Sciences issue’s scope.  

However, there are some unclear aspects related to the methods used. Some terms were not used, defined, or specified in the Method section. The author needs to include more detailed information about terms and indicate a specific research analysis process. In addition, some terms were used inconsistently. The author needs to clarify those terms and be consistent in using them. I have included detailed suggestions and questions below.

Also, throughout the paper, there were several typos and grammatical mistakes. Because of these errors, it was somewhat difficult to understand what the author wanted to tell. I recommend that the author proofreads his/her paper with a professional editor or translator.   

I used the following acronyms for review (L: Line, Q: Question, S: just suggestion, so edit at your discretion).

------------------------------------------------

L 30: Change “considering also” to “also considering”, Typo: delete additional period mark 

L 31: You mentioned that you conducted “a cross-sectional observational study”. I was not able to see any explanation about the observational study format in the Method section. Please add what observations you have conducted in your study and mention it in the Method section.

L 37: Typo: change “W” to “We”, You mentioned that you found “statistically significant negative correlation.” Please clarify the term “correlation.” In the method section, you briefly indicated that you used Multiple Linear Regression analysis. 

Did you try to find the correlation among the variables? If you have run an independent correlation test, please specify the process of the correlation test. If you just meant the simple relations among variables, please reword “correlation” to “relation.” Also look at Line 239,262, 324, 326

L 113: “260 healthy children” How do you know if these children are healthy or not? (Q) Please define “healthy children” and explain the logic of using “healthy” in the study.

L 121: Please indicate what acronym DD stands for. You are using this acronym for the first time, so you need to write “Digital Devie (DD).”

L 131: Please be consistent on a term. You used the “Digital Devic Questionnaire (DDQ)” on L 130, but you used “Digital Device questionnaire” on L 131.  

L 145: Please use a different term for “fall asleep”. 

You used a different term in Table 2. Be consistent with the term.

L 148: What is “recreational activities”? More detailed information would help the audience’s understanding.

L159: What is “age-weighted scores”? Please define the term and explain it.

L161: Please revise the sentence: “In the form “Words and Phrases” is considered only the LQ.”

L 160-161: What are AGQ, LUQ, LPQ, and LQ? (Q) You may need a definition and description of each term.  

L171: You need to explain what “other factors” are. Why did you select those specific variables? (Q) It would be helpful to the audience when you explain the logic behind the selection.

L176: “3.1 Sample characteristics” should be included in the Method section.

L 186: Change “F= Father” to “F= father”. Be consistent in capitalization.

L 189: You have the same subtitles for both 2.2 and 3.2. 

Please change one of the subtitles. (S) 

Use capital Q for the questionnaire. Be consistent in capitalization.

L199-200: From my perspective, this sentence is not necessary. 

If you wanted to identify the characteristics of groups, the mean differences could be explicitly added.

L207: “The children’s favorite DD contents” were videos with dialogue? Or “The children’s favorite DD” content was the video with dialogue? You need to fix the grammatical error here.

L216: taken off? Do you mean taken away?

Page 7 Table: Typo: Favorite contents- Videogames- Please delete “?” and add “)”  

L 232-234: You need to rewrite this sentence. Do you mean, “Subsequently, moderation analysis showed that other factors including sex, level of education of parents, the modality of use, and frequency of social activities do not have any influence on the result of your regressional analysis?” What does “Through subsequent moderation analyzes emerged that” mean? What is ““this relationship” in your sentence? If you would like to discuss non-significant variables, please reword and rephrase them.

L239: Please review the L37 recommendation and use the appropriate term.  

Table 3: Use consistent terms throughout the paper. 

In the Method section and Result write-up, you used “Language Understanding Quotient” and “Language Production Quotients”, but in Table 3, you used “Understanding Quotient” and “Production Quotient.”  

L262: Review L37 recommendation and clarify “correlate” 

L270: Consider using “97% of children” instead of using “almost all of the children” (S)  

L275: “for the Smartphone in the older” (?). Do you mean “in the older group”?

L284: Typo: include “to” after “important”. “important to underline”

L302: Typo: “the their”

L303: What does “A further 6%” mean?

L304: “taken away” or “turned off”?        

L304-306: Please clarify the terms and rewrite the sentence. What does “become totalizing” mean?

L324 & L326: 

Review L37 recommendation and clarify the term “correlattion” 

L353-363: Please make a paragraph compiling four sentences. 

You have four sentences that are four different paragraphs.  

L353: The expression “did not seem to” indicates too much uncertainty. Please use other words. (S)

L358-360: How does it work? You did not investigate older children’s direct social interactions. You need to have evidence of other research and proof of your data to back up this statement.

L360: Why did you put ‘direct’ here?  

L361: What is “indirect questionnaires”? Please define and explain it in the Method section. If you are going to use the term “indirect questionnaires” for DDQ and PVB, you need to indicate it in the Method section when you explained these two questionnaires. 

Author Response

Salerno, 17 September 2020

To Editor-in-Chief

Brain Sciences

Dear Editor,

Thank you for considering our manuscript for publication in Brain Sciences.

We greatly appreciated the reviewer’s comments and suggestions. We did our best to address all the issues raised by reviewer. The manuscript has been modified, according to the reviewer’s comments and remarks. We have italicized the comments and addressed each of them in a point-by-point reply.

We hope that the manuscript is now acceptable for publication in Brain Sciences.

Looking forward to hearing from you,

Grazia M.G. Pastorino

COMMENTS TO THE AUTHORS

REVIEWER: 2

Reviewer comment:

The study investigates the relationship between digital device use and language abilities in Italian children between the age of 8-36 months. The study’s significance would be an effort to provide valuable information on how the duration of using digital devices was related to children’s language skills. The study also provides a discussion of the negative effect of using digital devices in early years to parents and pediatricians. This study is interesting and significant in the field of language and literacy development of young children. The use of digital devices of young children is a very timely manner topic in this Pandemic situation since more digital devices are used in the education field. Thus, I believe that the findings reported in this study are relevant to the Brain Sciences issue’s scope.  

However, there are some unclear aspects related to the methods used. Some terms were not used, defined, or specified in the Method section. The author needs to include more detailed information about terms and indicate a specific research analysis process. In addition, some terms were used inconsistently. The author needs to clarify those terms and be consistent in using them. I have included detailed suggestions and questions below.

Also, throughout the paper, there were several typos and grammatical mistakes. Because of these errors, it was somewhat difficult to understand what the author wanted to tell. I recommend that the author proofreads his/her paper with a professional editor or translator.   

I used the following acronyms for review (L: Line, Q: Question, S: just suggestion, so edit at your discretion).

L 30: Change “considering also” to “also considering”, Typo: delete additional period mark 

L 31: You mentioned that you conducted “a cross-sectional observational study”. I was not able to see any explanation about the observational study format in the Method section. Please add what observations you have conducted in your study and mention it in the Method section.

L 37: Typo: change “W” to “We”, You mentioned that you found “statistically significant negative correlation.” Please clarify the term “correlation.” In the method section, you briefly indicated that you used Multiple Linear Regression analysis. 

Did you try to find the correlation among the variables? If you have run an independent correlation test, please specify the process of the correlation test. If you just meant the simple relations among variables, please reword “correlation” to “relation.” Also look at Line 239,262, 324, 326

L 113: “260 healthy children” How do you know if these children are healthy or not? (Q) Please define “healthy children” and explain the logic of using “healthy” in the study.

L 121: Please indicate what acronym DD stands for. You are using this acronym for the first time, so you need to write “Digital Devie (DD).”

L 131: Please be consistent on a term. You used the “Digital Devic Questionnaire (DDQ)” on L 130, but you used “Digital Device questionnaire” on L 131.  

L 145: Please use a different term for “fall asleep”. 

You used a different term in Table 2. Be consistent with the term.

L 148: What is “recreational activities”? More detailed information would help the audience’s understanding.

L159: What is “age-weighted scores”? Please define the term and explain it.

L161: Please revise the sentence: “In the form “Words and Phrases” is considered only the LQ.”

L 160-161: What are AGQ, LUQ, LPQ, and LQ? (Q) You may need a definition and description of each term.  

L171: You need to explain what “other factors” are. Why did you select those specific variables? (Q) It would be helpful to the audience when you explain the logic behind the selection.

L176: “3.1 Sample characteristics” should be included in the Method section.

L 186: Change “F= Father” to “F= father”. Be consistent in capitalization.

L 189: You have the same subtitles for both 2.2 and 3.2. 

Please change one of the subtitles. (S) 

Use capital Q for the questionnaire. Be consistent in capitalization.

L199-200: From my perspective, this sentence is not necessary. 

If you wanted to identify the characteristics of groups, the mean differences could be explicitly added.

L207: “The children’s favorite DD contents” were videos with dialogue? Or “The children’s favorite DD” content was the video with dialogue? You need to fix the grammatical error here.

L216: taken off? Do you mean taken away?

Page 7 Table: Typo: Favorite contents- Videogames- Please delete “?” and add “)”  

L 232-234: You need to rewrite this sentence. Do you mean, “Subsequently, moderation analysis showed that other factors including sex, level of education of parents, the modality of use, and frequency of social activities do not have any influence on the result of your regressional analysis?” What does “Through subsequent moderation analyzes emerged that” mean? What is ““this relationship” in your sentence? If you would like to discuss non-significant variables, please reword and rephrase them.

L239: Please review the L37 recommendation and use the appropriate term.  

Table 3: Use consistent terms throughout the paper. 

In the Method section and Result write-up, you used “Language Understanding Quotient” and “Language Production Quotients”, but in Table 3, you used “Understanding Quotient” and “Production Quotient.”  

L262: Review L37 recommendation and clarify “correlate” 

L270: Consider using “97% of children” instead of using “almost all of the children” (S)  

L275: “for the Smartphone in the older” (?). Do you mean “in the older group”?

L284: Typo: include “to” after “important”. “important to underline”

L302: Typo: “the their”

L303: What does “A further 6%” mean?

L304: “taken away” or “turned off”?        

L304-306: Please clarify the terms and rewrite the sentence. What does “become totalizing” mean?

L324 & L326:             

Review L37 recommendation and clarify the term “correlattion” 

L353-363: Please make a paragraph compiling four sentences. 

You have four sentences that are four different paragraphs.  

L353: The expression “did not seem to” indicates too much uncertainty. Please use other words. (S)

L358-360: How does it work? You did not investigate older children’s direct social interactions. You need to have evidence of other research and proof of your data to back up this statement.

L360: Why did you put ‘direct’ here?  

L361: What is “indirect questionnaires”? Please define and explain it in the Method section. If you are going to use the term “indirect questionnaires” for DDQ and PVB, you need to indicate it in the Method section when you explained these two questionnaires. 

Authors’ response:

We thank the reviewers for all the comments and suggestions. We made all required changes, as follows:

L 30: We change “considering also” to “also considering” and we deleted additional period mark, as suggested (line 30).

L 31: As suggested, we explained the format of the observational study both in the Abstract and in the Methods section (line 31 and line 114), modifying the sentence as follows: "We conducted a cross-sectional observational study on digital devices use and language abilities in 260 children (140 males = 54%) aged between 8-36 months (mean = 23.5 ± 7.18 months)"

L 37: We changed “W” to “We”, as suggested (line 37). We have clarified the statistical analysis used in the Abstract “Linear regression analysis was used to evaluate the relation between different variables” (lines 35-36) and in the specific section of the Methods “Linear regression analysis was used to evaluate the relation between different variables” (lines 195-196), specifying that we have used linear regression analysis to investigate the relationship between use of digital devices and language. Consequently we have replaced the term "correlation" with the term "relation" throughout the manuscript.

L 113: we have eliminated the word "healthy" because it is ambiguous both in the abstract (line 32) and in the Methods (line 115). We specified in the exclusion criteria that all children recruited were not affected by neuropsychiatric conditions: “the exclusion criteria were the presence of medical or neuropsychiatric conditions that could affect language or neuropsychomotor development, and poor parental compliance to take parte into the study.” (lines 117-119).

L 121: we made this change by adding the abbreviation DD after the first time we mentioned Digital Devices, at the end of the Introduction section, (line 102).

L 131: we changed the capital letter as suggested (line 149)

L 145: we replaced the term “to fall asleep” with “to let the child sleep” (line 164 and Table 2)

L 148: we replaced “recreational activities” with “social activities” and we added the following clarification: “(times in which the child plays or interacts with other children)” (line 167-168)

L159: we replaced “age-weighted scores” with “scores standardized for age” (line 179)

L161: we rephrased the sentence “In the form “Words and Phrases” is considered only the LQ.” as follow: “In the "Words and Phrases" Form, one standardized score is considered: Lexical Quotient (LQ): parameter that evaluates the production of words”(lines 186-187).
L 160-161: we defined all the Quotients, as follow (lines 181-187): 

  • Actions and Gestures Quotient (AGQ): parameter that evaluates the mimic-gestural communication skills.
  • Lexical Understanding Quotient (LUQ): parameter that evaluates the comprehension of words.
  • Lexical Production Quotient (LPQ): parameter that evaluates the production of words.
  • Lexical Quotient (LQ): parameter that evaluates the production of words.

L171: we specified what the others factors we considered are: gender, parents’ educational level, parent’s job, co-viewing, contents of DD, frequency of social activities (line 197-198) in the Method section.
We explained the raisons of our choise in the Introduction section “The secondary objective of the study was to evaluate the influence of other factors on this relationship. Based on previous literature data, the following variables were selected: gender, socio-economic status, co-viewing, contents of DD, frequency of social activities [5-6,9-11]. We hypothesize that co-viewing and type of content of DD influence the relationship between DD use and language development. In particular, we consider that there is a difference between children that use devices in active interaction with parent and children that use devices alone. We do not expect socioeconomic status differences as each child has full access to devices” (lines 105-110).
L176: We included the sample characteristics in the Method section (lines 135-141) “2.1 Sample selection and characteristics
L 186:  we changed “F= Father” to “F= father” in table 1 (line 145).
L 189: we changes the sequent subtitles “3.1 Use of Digital Devices by children” (line 203).
L199-200: we added all the mean differences as follow: (Smartphone: 11.1±3.1 vs 16.4±5.9; Tablet: 15.3±3.1 vs 18.6±6.2; Personal Computer: 14.0±3.1 vs 16.8±12.9; Television: 8.4±3.7 vs 12.2±5.3; Videogames: 16.0±0.0 vs 23.6±1.3) (lines 215-216)
L207: we corrected the phrase in “The children’s favorite DD content was the video with dialogue” (line 223)
L216:  we modified “taken off” with “taken away” (lines 232 and 320)
Table 2: we delete “?” and add “)” 
L 232-234: we rephrased the sentence as follow “Subsequently, moderation analysis showed that other factors including gender, level of education and job of parents, co-viewing, modality of DD use, and frequency of social activities do not have a significant influence on the result of the regression analysis.” (lines 247-249)
L239: we replaced “correlation” with “relation” (line 255)  
Table 3: we modified Table 3 using the following terms: “Lessical Understanding Quotient”, “Lessical Production Quotient” and “Lessical Quotient”. We renamed the names of the Quotients in the manuscript as well.
L262: we replaced “correlate them” with “relate them” (line 278)  
L270: we changed sentence as follow: “… was that 97% of children used at least one” (lines 285)
L275: we modified “for the Smartphone in the older” in “for the Smartphone in the older group” (line 291)
L284: we corrected the typo, adding “to” in the sentency “important to underline” (line 300)
L302: we corrected the typo, deleting “the” in the sentency “the their” (line 318)
L303: we modified the phrase “A further 6%” mean” in “6% of the children…” (line 319)
L304: we changed in “taken off” in “taken away” (line 320).
L304-306: we eliminated “become totalizing” and we modified the sentency as follow “This data suggests that the use of digital devices in some children can lead to a reduction in social interaction and a difficulty in emotional regulation, as already reported in previous studies” (lines 320-322).
L324 & L326: we replaced “correlation” with “relation” (line 340 and 342).  
L353-363: we made a single paragraph, as suggested (lines 369-374).
L353: we changed the expression “did not seem to” in “did not significantly affect “ (line 369)
L358-360: We delated this sentency.
L360: We delated this word.
L361: We deleted the term “indirect” because we meant “questionnaires aimed at parents”, as already explained in the Methods section (line 373).
